# Combined Radionuclide Therapy and Immunotherapy for Treatment of Triple Negative Breast Cancer

**DOI:** 10.3390/ijms22094843

**Published:** 2021-05-03

**Authors:** Alyssa Vito, Stephanie Rathmann, Natalie Mercanti, Nader El-Sayes, Karen Mossman, John Valliant

**Affiliations:** 1Department of Medicine, McMaster Immunology Research Centre, McMaster University, Hamilton, ON L8S 4K1, Canada; vitoar@mcmaster.ca (A.V.); elsayesn@mcmaster.ca (N.E.-S.); 2Department of Chemistry and Chemical Biology, McMaster University, Hamilton, ON L8S 4K1, Canada; rathmasm@gmail.com (S.R.); mercanne@mcmaster.ca (N.M.)

**Keywords:** triple negative breast cancer, immunotherapy, immune checkpoint therapy, radionuclide therapy

## Abstract

Triple negative breast cancer (TNBC) is an aggressive subtype of the disease with poor clinical outcomes and limited therapeutic options. Immune checkpoint blockade (CP) has surged to the forefront of cancer therapies with widespread clinical success in a variety of cancer types. However, the percentage of TNBC patients that benefit from CP as a monotherapy is low, and clinical trials have shown the need for combined therapeutic modalities. Specifically, there has been interest in combining CP therapy with radiation therapy where clinical studies primarily with external beam have suggested their therapeutic synergy, contributing to the development of anti-tumor immunity. Here, we have developed a therapeutic platform combining radionuclide therapy (RT) and immunotherapy utilizing a radiolabeled biomolecule and CP in an E0771 murine TNBC tumor model. Survival studies show that while neither monotherapy is able to improve therapeutic outcomes, the combination of RT + CP extended overall survival. Histologic analysis showed that RT + CP increased necrotic tissue within the tumor and decreased levels of F4/80+ macrophages. Flow cytometry analysis of the peripheral blood also showed that RT + CP suppressed macrophages and myeloid-derived suppressive cells, both of which actively contribute to immune escape and tumor relapse.

## 1. Introduction

Breast cancer is the most common cancer among women worldwide, accounting for more than two million new cases and 600,000 deaths annually [1]. Triple negative breast cancer (TNBC) accounts for 10–20% of all breast cancers, presents a higher risk in women under the age of 40, demonstrates substantial tumor heterogeneity, and is often identified as being high grade [2]. TNBC patients routinely undergo extensive, highly toxic treatment regimens and have the highest risk of relapse amongst all breast cancer types [3,4]. Furthermore, recently approved therapies for TNBC are limited (olaparib, atezolizumab, and sacituzumab–govitecan) and only benefit 10–20% of patients, highlighting the need for improved therapies for TNBC patients. To this end, a deeper understanding of the immune landscape in TNBC patients is required to develop novel, effective therapies.

Recent years have seen the emergence of immunotherapies in both preclinical and clinical development, revolutionizing the way we think about treating cancer patients. One such therapy, immune checkpoint blockade (CP), uses antibodies to block inhibitory pathways on immune cells and has shown widespread clinical success with durable cures across a variety of cancer types [5,6]. However, the percentage of patients that respond to CP is low, and even those patients who initially display tumor regression often succumb to relapsed disease [5,7]. As the field of immuno-oncology continues to grow, so too does our understanding of immunotherapies and the challenges associated with achieving durable and complete responses to treatment. In an effort to combat clinical barriers to CP efficacy, there has been an emergence of new paradigms incorporating traditional therapies into immunotherapy regimens [8,9,10].

Radiation therapy has been a mainstay treatment for many forms of cancer since the late 1800s. Historically, radiation has been thought to work solely through direct contact-based killing, but there has long been the postulation of immune involvement through the hypothesis of the abscopal effect [11]. The abscopal effect occurs when an irradiated tumor initiates a cascade of events with the release of damage-associated molecular patterns (DAMPs) such as high mobility group box 1 (HMGB1), adenosine triphosphate (ATP), and heat shock proteins (HSPs). These DAMPs act on receptors that are expressed on dendritic cells (DCs), leading to antigen presentation, tumor-specific killing from cytotoxic T cells, and ultimately anti-tumor activity (Figure 1). The abscopal effect was first clinically documented in 1953 [12], but interest waned with rare occurrences noted and difficulties in recapitulating the phenomenon in preclinical models. Now, in the era of immunotherapy, there is a much deeper understanding of the immune system and the interplay of cells in the tumor microenvironment (TME), and the abscopal effect has once again been brought to the forefront of oncologic research.

Clinical studies have shown that the combination of radiotherapy and immunotherapy synergizes for enhanced anti-tumor activity and improved prognostic outcomes [13,14,15]. In a phase I clinical trial, patients with metastatic or nonresectable melanoma tumors were treated with anti-CTLA-4 and anti-PD-1 checkpoint antibodies, and a small cohort of patients was also given radiation therapy [15]. This study noted a significant improvement in overall survival percentages in the patient population receiving the dual therapy. Similarly, in a phase II clinical trial, patients with metastatic TNBC (mTNBC) were treated with anti-PD-1 checkpoint antibodies and fractionated external beam radiation therapy [16]. The overall response rate of the patients was 17.6% with three responders achieving a 100% reduction in tumor volume outside of the irradiated field. It is important to note that the patients enrolled in the study were unselected for PD-L1 expression and had failed previous first-line therapies.

Here, we outline a therapeutic platform using cytotoxic radiation to sensitize otherwise non-responsive tumors to CP. In particular, we chose to investigate whether continuous cell irradiation, through the use of internal radionuclide therapy (RT), would synergize with dual anti-CTLA-4 and anti-PD-L1 checkpoint therapy. In order to maximize and better control the radiation dose to the tumor, we used intratumoral delivery of the beta emitter, lutetium-177, linked to a biomolecule. Albumin, an abundant blood protein, was used as a biocompatible protein anchor to prolong retention of the radionuclide in the tumor. With respective to potential translation, intratumoral injections of albumin radiopharmaceuticals for sentinel node imaging in breast cancer are routinely performed in the clinic [17], while materials labeled with beta emitting isotopes are routinely being delivered in a similar manner for liver cancers.

## 2. Results

### 2.1. Tunable Platform for Intratumoral Administration of Radiotherapeutic

Due to the rich history and proven track record in medical practice, we chose to use albumin as the protein anchor to prolong the retention of the radionuclide within the TME and provide greater control over the administered dose when compared to intravenous (iv) administration [18,19,20]. To increase the versatility of our platform and ensure that protein integrity is maintained during synthesis, bovine serum albumin (BSA) was first functionalized with trans-cyclooctene (TCO). Then, this TCO–BSA conjugate can undergo a rapid, room temperature inverse electron-demand Diels–Alder (IEDDA) reaction in which the TCO moiety forms a covalent linkage with its coupling partner, a tetrazine, which in this case is radiolabeled with lutetium-177 (Figure 2). This type of two-step functionalization minimizes the risk of a non-specific binding interaction of the radionuclide with BSA. To prepare TCO–BSA, BSA was combined with a TCO–NHS ester, and the mixture was allowed to incubate overnight at room temperature. TCO–BSA was purified by dialysis, and the conjugation was confirmed by matrix-assisted laser desorption ionization mass spectrometry (MALDI-MS; Appendix A). The tetrazine was synthesized as previously described [21], and the radiolabeled product was produced by adding [177Lu]LuCl3 (Figure 3) at 60 °C for 5 min, resulting in a radiochemical yield of >99%. The radiolabeled small molecule was incubated with TCO–BSA for ten minutes at room temperature, followed by purification using a high molecular weight spin filter. The resultant radiochemical yield was 46 ± 5% based on the amount of activity isolated from the spin filter.

### 2.2. RT Immobilizes in the Tumor Microenvironment

To evaluate the spatial distribution of the compound within the tumor, qualitative autoradiography studies were performed. Subcutaneous E0771 tumors were grown in C57/Bl6 mice. Mice were treated with a single intratumoral injection of RT when palpable tumors arose (12 days after implantation), and groups of mice were sacrificed at 24, 72, and 120 h (Figure 4). These images revealed that the compound was able to distribute well throughout the tumor after a single injection, which is evident out to 120 h. To quantitatively evaluate the long-term retention of the RT in the tumor as well as to assess uptake in non-tumor tissues, biodistribution studies were performed with a direct comparison between intratumoral and intravenous administration. RT delivered intratumorally showed high retention in the tumor out to 120 h as well as high tumor to non-tumor ratios, which are ideal for therapy (Appendix A, n = 3). As expected, intravenous administration of the RT resulted in poor tumor retention and high uptake in non-tumor tissues (Appendix A, n = 3).

### 2.3. Radiotherapy Results in Improved Prognostic Outcomes

While radionuclide therapy may be administered as a single or fractionated dose in the clinic, studies suggest improved therapeutic efficacy and enhanced antitumor immunity with fractioned regimens, employing intratumoral injections of as little as 2 MBq per dose in murine xenograft models [22,23,24,25]. In an effort to determine the most efficacious dosing regimen for our RT, dose optimization studies were performed. C57/Bl6 mice bearing E0771 subcutaneous tumors on the left flank were treated with RT intratumorally and monitored for overall survival. Preliminary studies utilized a two-dose regimen, with single doses up to 0.74 MBq (Appendix A). This range of radioactivity was insufficient to slow tumor progression, and therefore, the maximum dose was increased to 3.33 MBq and administered as a single dose to monitor host toxicity and tolerability (Appendix A). Mice tolerated treatment well with no acute toxicity seen. Moving to multi-dosing to promote improved efficacy and sustained tumor regression, two doses, with doses ranging from 0 to 4.44 MBq, were given five days apart. This regimen resulted in delayed tumor progression and improved survival outcome (Figure 5).

### 2.4. RT + CP Improves Overall Survival in Tumor-Bearing Mice

As CP continues to gain traction as a viable therapeutic option, using PD-L1 expression as a predictive biomarker has become more commonplace. TNBC tumors do indeed express PD-L1 [26,27]; however, the expression is low, and it is not homogenously distributed throughout the tumor, but rather found in focal areas in a small proportion of cancer cells [28]. Further to this, clinical trials have reported both the efficiency and necessity of combined therapeutic modalities, as TNBC patients often have short-lived responses to CP on its own [29]. Based on our preliminary studies, we hypothesized that high-dose RT is capable of sensitizing tumors to CP. Survival studies were performed with the addition of dual CP targeting the non-redundant pathways of cytotoxic T-lymphocyte antigen 4 and programmed death ligand-1 (with anti-CTLA4 and anti-PD-L1 antibodies, respectively). While neither control (non-radioactive TCO-BSA), RT alone, CP alone, or vehicle control + CP showed therapeutic efficacy, the combination of RT + CP resulted in greatly improved overall survival (Figure 6). Mice tolerated treatments well with no toxicity seen.

### 2.5. Radiotherapy + CP Increases TILs in Otherwise Immune-Bare Tumors

To further investigate the impact of each component of our therapy to the TME, histologic assessment was performed. Tumors were harvested on day 7 from mice treated with PBS, control, CP, RT, control + CP, and RT + CP. Analysis of whole tumor sections harvested and stained with hematoxylin and eosin (H&E) shows that mice treated with PBS or the control compound have large tumors with many multi-nucleated cells, suggesting rapid cellular division. Mice treated with CP, RT, or control + CP present with pockets of necrosis and many multi-nucleated cells surrounding these areas, suggesting that although these therapies may induce acute necrosis in areas of the tumor, these mice still have rapid tumor kinetics and disease progression. As expected from survival study outcomes, tumors harvested from mice that were treated with RT + CP present with increased necrosis and shrinking cellular structures, which was likely a direct result of their response to therapy.

Tumors were further stained with CD4, CD8, and F4/80 to assess immune cell infiltrates in the tumor. Mice treated with PBS or the control compound present with moderate levels of CD4^+^ and CD8^+^ cells with densely populated areas of F4/80^+^ macrophages. Mice treated with CP therapy have increased levels of CD4^+^ and CD8^+^ cells and substantial increases of F4/80^+^ macrophages. Interestingly, mice that were treated with RT alone have significantly decreased levels of CD4^+^ and CD8^+^ cells, with a moderate decrease in F4/80^+^ macrophages. Control + CP tumors appeared very similar to those treated with CP alone, suggesting that CP was able to increase the level of CD4^+^, CD8^+^, and F4/80^+^ cells in the tumor. Mice treated with RT + CP have moderate levels of CD8^+^ T cells (similar to PBS treated mice), decreased levels of F4/80^+^ macrophages, and decreased levels of CD4^+^ T cells.

### 2.6. RT + CP Decreases Immunosuppressive MDSCs in the Peripheral Blood

To investigate the systemic effects of therapeutic intervention, we performed immune analysis studies. E0771 tumors were grown in C57/Bl6 mice and treated with PBS, control, CP, RT, control + CP, and RT + CP. RT doses were kept consistent with previous studies in Figure 6 and Figure 7. Blood was drawn on days 4 and 9, and peripheral blood mononuclear cells (PBMCs) were analyzed via flow cytometry (Figure 8). While no significant difference was seen in CD4^+^ or CD8^+^ T cells, RT and RT + CP significantly reduced macrophages (F4/80^+^ cells), B cells (B220^+^ cells), and dendritic cells (DCs, CD11c^+^ cells) on day 4. Interestingly, while no therapy was able to suppress myeloid-derived suppressor cells (MDSCs; Ly6G^hi^Ly6C^int^ cells) at day 4, both control + CP and RT + CP significantly decreased the frequency of circulating MDSCs on day 9. Again, we see that B cells were suppressed by RT and RT + CP at day 9, which we believe can be attributed to their high level of radiosensitivity [30].

## 3. Discussion

Clinical studies have detailed the synergistic benefit of combined external beam radiation therapy and immunotherapy [31,32,33,34,35,36]. Hodge and colleagues demonstrated that external beam radiation of tumors can alter tumor phenotype, rendering it susceptible to immune-mediated killing [37]. While external beam radiation has been a mainstay, first-line therapy for many types of cancer for more than 100 years, it comes with unfavorable side effects such as damage to surrounding tissues and limited utility in metastatic disease. To combat this, researchers have shifted toward the development of internal RT, which can deliver cytotoxic levels of radiation directly to disease sites with a high level of specificity. Indeed, RT can be highly selective not only due to the nature of the targeting vector chosen but also through the choice of radionuclide used. For example, beta-emitting radioisotopes are unable to drive therapeutic response in hypoxic tumors [38,39]. In such an instance, the radionuclide can be changed to an alpha emitter, such as actinium-225, for improved efficacy and decreased toxicity to the patient. This allows for the development and utilization of a single tunable probe, which can then be personalized for optimal effectiveness for individual cancers.

In this paper, we have investigated the combined effects of RT with the beta-emitting radionuclide lutetium-177 and CP immunotherapy using anti-PD-L1 and anti-CTLA-4 in an E0771 murine TNBC tumor model. While dual checkpoint blockade has resulted in increased toxicities in patients in clinical trials [40,41], our mice tolerated this treatment very well, with no toxicity seen with CP or combination regimens. CP alone showed no survival benefit in our model. The combination of control + CP did show modest benefit, which was likely due to increased T cell-mediated killing from CP therapy. However, this benefit was not translated into significant improvements in overall survival, which was only seen with the combination of RT + CP. While studies have documented correlations between increased levels of cytotoxic T cells and improved overall survival in TNBC patients [42], this alone is often insufficient to overcome intrinsic resistance mechanisms and tumor relapse. Indeed, resistance mechanisms are most commonly driven by the immunosuppressive TME, where MDSCs, regulatory T cells (Tregs), and tumor-associated macrophages (TAMs) play a crucial role. Indeed, Tregs are a strong prognostic predictor of therapeutic outcome in TNBC patients [43]. IHC analysis on our treated tumors showed that RT + CP is able to decrease TAMs and CD4^+^ T cells. While we did not phenotype the CD4^+^ T cell population to show they are specifically Tregs, this therapeutic combination has virtually depleted all CD4+ cells, indicating a direct effect on decreasing Tregs (which are always CD4^+^).

MDSC frequency is directly correlated with tumor progression, recurrence, poor prognosis, and decreased efficacy of immunotherapies [44]. In our therapeutic combination, we have shown that RT + CP is able to suppress the frequency of MDSCs in peripheral blood, potentially contributing to the observed improved therapeutic efficacy in the form of increased survival. While we do also see the suppression of MDSCs in the control + CP group, this does not correlate with a benefit in overall survival and may simply be the effect of increase T cell killing due to CP administration. Additionally, in TNBC, TAMs have been shown to promote tumor growth and progression while also modulating the levels of PD-L1 expression [45]. In our studies, we have shown that both RT and RT + CP suppress peripheral macrophages as early as day 4 (Figure 8), suggesting that early treatment with RT aids in alleviating TAM modulation of PD-L1 suppressive functions, although more experiments are required to properly investigate this phenomenon.

Immunotherapies aim to stimulate the immune system to mount a systemic anti-tumor immune response to recognize and destroy tumor cells within the body. Consideration of the abscopal effect and the possibility that RT can truly induce a bona fide anti-tumor immune response greatly expands the breadth of application for this therapeutic platform, as it need not be used solely for primary lesions but can also induce the regression of distant microscopic lesions as well. While this work was completed in a transplantable murine model of TNBC, these studies can be applied to many solid tumor types, increasing the potential translatability of our findings. Additionally, the use of non-radiolabeled BSA was shown to have no influence on therapeutic outcomes or tumor kinetics (as assessed with our control groups). For clinical translation, the BSA derivative can be replaced with the corresponding human serum albumin analogue.

Our studies are limited by the nature of murine hosts and their inability to accurately represent human biology. Indeed, cancer metastasis is a major cause of failed therapeutic intervention and cancer-related deaths [46,47], and our data were conducted in a subcutaneous tumor model representative of primary tumor formation. For enhanced translative capacity, our therapeutic platform should be studied in metastatic and spontaneously arising tumor models to better recapitulate de novo tumor formation in a host. In these models, we would treat the primary tumor and monitor response in metastatic lesions in terms of both the size and number of lesions formed.

## 4. Materials and Methods

### 4.1. Cell Lines

Murine medullary breast adenocarcinoma cells isolated as a spontaneous tumor from a C57/Bl6 mouse (E0771; CH3 Biosystems, Amherst, NY, USA) were maintained in Roswell Park Memorial Institute (RPMI) medium supplemented with 10% FBS, 10 mM HEPES, 200 μM geneticin, and 2 mM L-glutamine. Murine mammary gland breast cancer cells isolated as a spontaneous tumor from a Balb/c mouse (4T1; ATCC^®^ CRL2539™) were maintained in RPMI medium supplemented with 10% FBS, 2 mM L-glutamine, 100U/mL penicillin and 100 μg/mL streptomycin. All cells were grown at 37 °C with 5% CO_2_.

### 4.2. Chemistry General

Chemicals and reagents for synthesis were purchased from Sigma-Aldrich and Conjuprobe and used without further purification. ^177^Lu[Lu] was produced by the McMaster Nuclear Reactor (MNR, Hamilton, Ontario, Canada) using the ^176^Lu (p,γ) reaction and was provided as a solution of [^177^Lu]LuCl_3_ in 0.01 M HCl. Radio-TLC was performed using a Bioscan AR-2000 imaging scanner (West Vancouver, BC, Canada) on iTLC-SG glass microfiber chromatography paper (SGI0001, Agilent Technologies, Santa Clara, CA, USA) plates using 0.1 M EDTA as the eluent. For each TLC performed, plates were spotted with approximately 2 μL (≈3.7 kBq) and run for 5 min. MALDI data were obtained using a Bruker Ultraflextreme spectrometer (Billerica, MA, USA).

### 4.3. In Vivo Therapy Experiments

Mice were maintained at the McMaster University Central Animal Facility, and all the procedures were performed in full compliance with the Canadian Council on Animal Care and approved by the Animal Research Ethics Board (Animal Utilization Protocol 17-05-22, January 2020) and the Health Physics Department of McMaster University (Permit KM-1, October 2018). Six- to eight-week-old female C57/Bl6 mice (Charles River Laboratories, Wilmington, MA, USA) were used to implant 5 × 10^6^ E0771 cells subcutaneously on the left flank. Mice were weighed and all were found to be approximately 20 g in size. Mice were housed in groups, 5/cage, fed a normal diet, and kept at room temperature. To minimize experimental variability, low-passage E0771 cells were used for subcutaneous injections. Twelve days after injection, the tumors reached treatable average tumor volume (50–100 mm^3^). Mice were blindly randomized prior to the start of treatment, but not blinded once treatments commenced. In experimental groups receiving control treatment, mice were treated on day 1 and day 5 with DNP-DOTA-BSA (100 µg/50 µL PBS, intratumorally). Experimental groups receiving RT treatment were treated on day 1 and day 5 with ≈4.44 MBq of ^177^Lu-DNP-DOTA-BSA (100 µg/50 µL PBS, intratumorally). Experimental groups receiving CP were treated with α-CTLA-4 (BioXCell, BE0131) and α-PD-L1 (BioXCell, BE101) antibodies (200 µg/200 μL PBS each, intraperitoneally) starting on day 3, every 3 days until mice reached endpoint or a total of 10 doses had been given. For all mouse studies, tumors were measured every 2–3 days, and mice having a tumor volume of 1000 mm^3^ were classified as endpoint.

### 4.4. Radiochemistry Methods

To a solution of tetrazine, a small molecule (100 µg, 48.0 nmol) in 100 µL of 0.1 M NaOAc (pH 5.5) was added [^177^Lu]LuCl_3_ (31–74 MBq). The reaction mixture was heated to 60 °C for 5 min, at which point a radio-TLC (cellulose/silica plate) was run in 0.1 M EDTA solution. The radiochemical yield of the reaction was determined to be >99% with >99% radiochemical purity. The radiolabelled tetrazine was added to a solution of TCO-BSA (2 mg/mL) in saline at room temperature for 10 min. The reaction was added to a 50 kDa spin filter and centrifuged at 4000 rpm for 10 min, which had been previously activated with 1.00 mL of saline. The supernatant was washed twice with 1 mL of sterile saline and centrifuged as stated above, which was followed by resuspension in sterile saline for injection. The conjugation efficiency of the reaction was 46 ± 5% (n = 3).

### 4.5. Autoradiography

C57Bl/6 mice bearing an E0771 flank tumor were administered a single dose of radiolabeled BSA (0.15–0.33 MBq/100 µg, intratumorally) on day 12 of growth when the tumors were palpable (≈100 mm^3^). The mice were sacrificed after 24, 72, or 120 h (n = 3), at which point the tumors were harvested, placed on a cryomold, and submerged in optimal cutting temperature compound. Then, the cryomold was wrapped in plastic wrap and flash frozen in liquid nitrogen for 15 s. The tumors were sent for analysis, where they were sliced and placed on a phosphor screen for 10 days.

### 4.6. Biodistribution Studies

Female, 5–6-week-old Balb/c mice ordered from Charles River Laboratory (Kingston, NY) were inoculated with 1 × 10^6^ 4T1 breast cancer cells in the right flank. On day 7 of growth, the mice were administered radiolabeled BSA (0.07–0.33 MBq/100 µg, intratumorally). At 72 and 120 h post-injection (n = 3 per time point), mice were anesthetized with 3% isoflurane and euthanized by cervical dislocation. Blood, adipose, adrenals, bone, brain, gall bladder, heart, kidneys, large intestine and caecum (with contents), liver, lungs, pancreas, skeletal muscle, small intestine (with contents), spleen, stomach (with contents), thyroid/trachea, urine + bladder, tumor, and tail were collected, weighed, and counted in a gamma counter. Decay correction was used to normalize organ activity measurements to time of dose preparation for data calculations with respect to injected dose (i.e., %ID/g).

### 4.7. Histology

Non-radioactive tumors were resected on day 7, fixed in 10% formalin for 48 h, and then transferred to 70% ethanol for immediate histological processing. Radioactive tumors were resected on day 7, fixed in 10% formalin, decayed for 3 months, and then transferred to 70% ethanol for histological processing. Tumor tissue was embedded in paraffin, and 4-μm sections were prepared. Tissue sections were processed for hematoxylin staining and IHC using Automated Leica Bond Rx stainer with Bond Refine Polymer Detection kit (Leica, DS9800). All antibodies were diluted in IHC/ISH Super Blocker (Leica, PV6199). Primary antibodies and working dilutions using HIER Retrieval Buffer 2 (Leica, AR9640) were as follows: CD3 (1:150; Abcam, ab16669), CD4 (1:800; eBio, 14-9766), and CD8a (1:1000; eBio, 14-0808). For F4/80 (1:500; AbD Serotec, MCA497R), an Enzyme 1 pre-treatment was performed before staining with antibody (AR9551). Antibodies CD4, CD8a, CD19, and F4/80 all required a secondary antibody before polymer detection using Rabbit anti-rat (Vector labs BA4001) at a dilution of 1:100. Immunohistochemistry slides were digitalized using the Olympus VS120-L100-W automated slide scanner (Shinjuku, Tokyo, Japan). They were batch-scanned on the brightfield setting at 20× magnification. The color camera used was the Pike 505C VC50.

### 4.8. Flow Cytometry Analysis

First, 150 µL blood was collected from the periorbital sinus. Red blood cells from all samples were lysed using ACK buffer. The PBMCs were treated with anti-CD16/CD32 (Fc block) and surface stained with fluorescently conjugated antibodies for FVS (BD Biosciences, #564406), CD4 (BD Biosciences, #561830), CD8 (BD Biosciences, #563046), CD11b (BD Biosciences, #553311), Ly6C (BD Biosciences, #553104), Ly6G (BD Biosciences, #560602), and F4/80 (BD Biosciences, #743282). An LSRFortessa flow cytometer (BD Biosciences, Mississauga, ON, Canada) with FACSDiva software (BD Biosciences, Mississauga, ON, Canada)) was used for data acquisition and FlowJo (Ashland, OR, USA) Mac, version 10.0 software was used for data analysis.

### 4.9. Statistical Analysis

For each statistical analysis used, normality of the distributions and variance assumptions were tested before running the statistical analyses. Multiple t-tests were used to determine the statistical significance of the differences in means. The log-rank Mantel–Cox test and the Gehan–Breslow–Wilcoxon test were used to determine statistical significance for the difference in Kaplan–Meier survival curves between treatments. All the tests were two-sided. The null hypothesis was rejected for p-values less than 0.05. All data analyses were carried out using GraphPad Prism (San Diego, CA, USA).

## Figures and Tables

**Figure 1 ijms-22-04843-f001:**
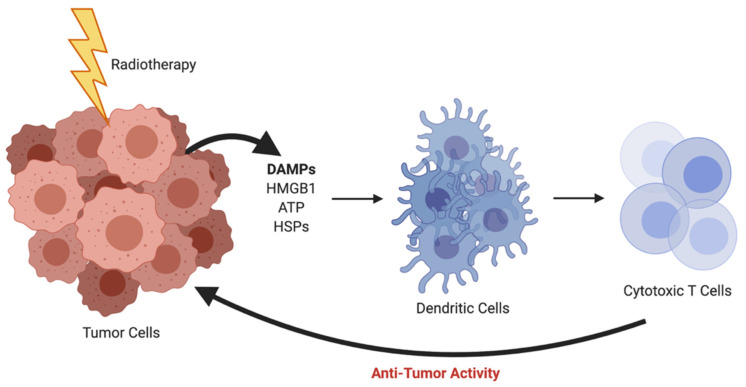
Radiotherapy-induced abscopal effect resulting in anti-tumor activity. *Created using BioRender.com.

**Figure 2 ijms-22-04843-f002:**
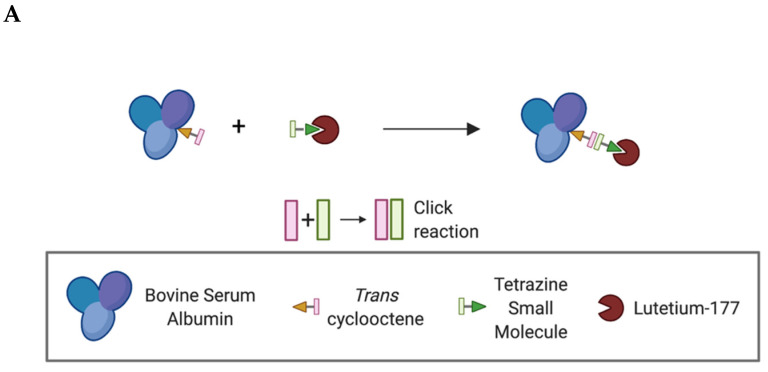
Schematic representation showing the inverse electron-demand Diels–Alder reaction between the trans-cyclooctene and tetrazine-based moieties. (**A**) Simplified schematic of the overall conjugation and labeling strategy. *Created using BioRender.com. (**B**) Schematic showing the key functional groups used to label albumin.

**Figure 3 ijms-22-04843-f003:**
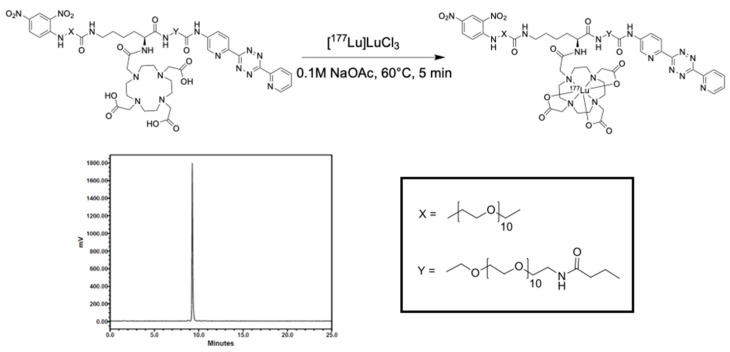
Radiolabeling scheme and radio-HPLC chromatogram of the tetrazine small molecule labeled with lutetium-177.

**Figure 4 ijms-22-04843-f004:**
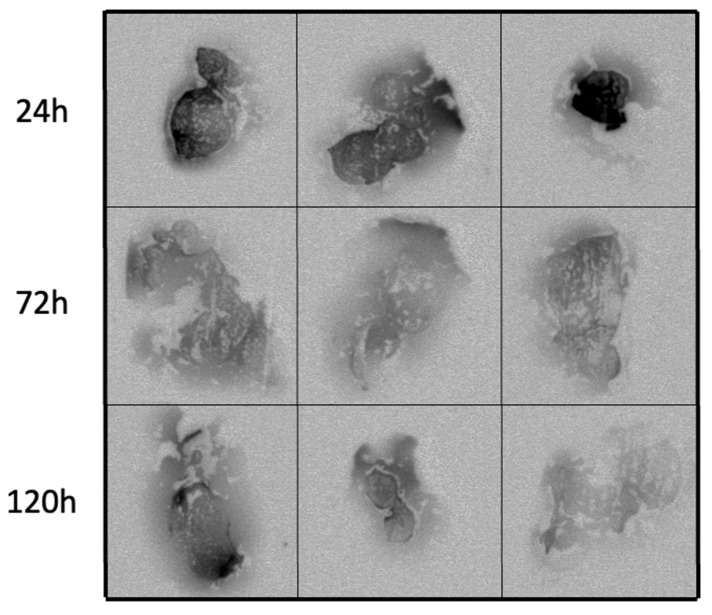
Autoradiography of tumors treated with a single intratumoral injection of RT shows distribution in the TME. C57/Bl6 mice bearing subcutaneous E0771 tumors were treated with a single intratumoral dose of RT (0.15–0.30 MBq). Mice were sacrificed at 24, 72, and 120 h after treatment, and tumors were harvested and flash frozen for autoradiography. Each image represents a slice of an individual tumor. The darker the area, the more radioactive decay that was detected in that area.

**Figure 5 ijms-22-04843-f005:**
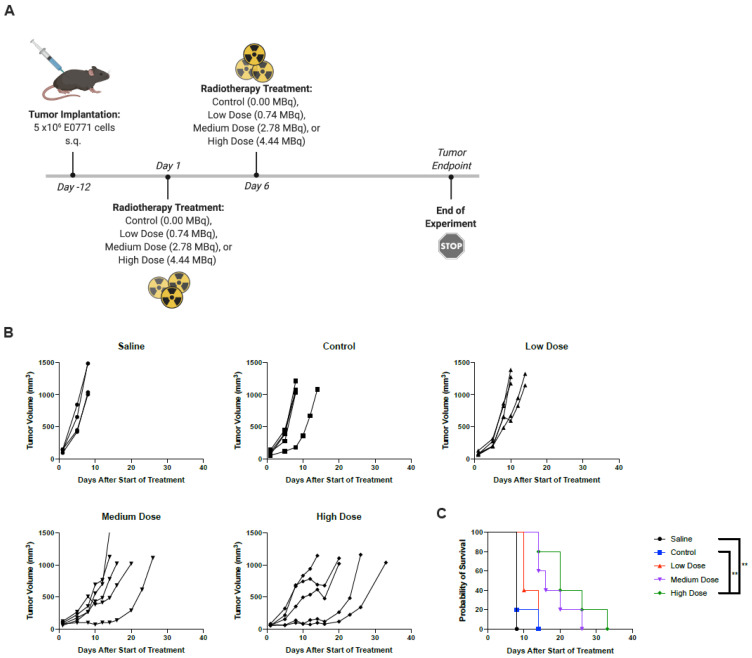
Two administrations of the highest dose RT improved survival outcomes. (**A**) C57/Bl6 mice bearing subcutaneous E0771 tumors on the left flank were treated with PBS, control (non-radioactive compound), low dose (0.74 MBq), medium dose (2.78 MBq), or high dose (4.44 MBq) on days 1 and 6. *Created using BioRender.com. (**B**) Tumor volumes were measured every 2–3 days from the start of treatment until mice reached endpoint. Each line represents an individual mouse within the group. (**C**) Kaplan–Meier survival curves of each group. ** *p* < 0.01.

**Figure 6 ijms-22-04843-f006:**
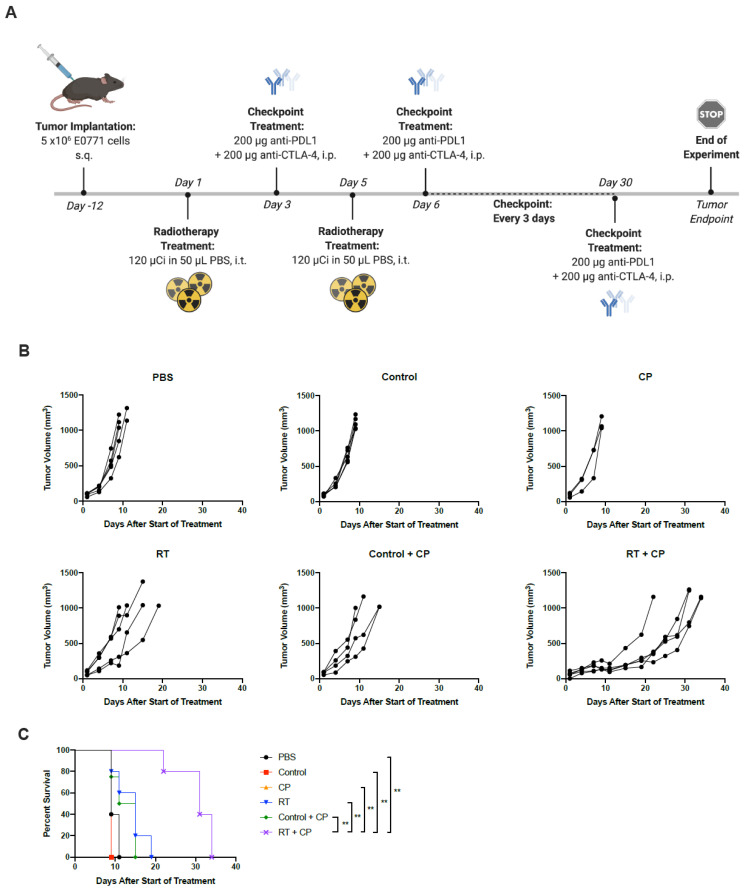
RT + CP significantly improves overall survival. (**A**) C57/Bl6 mice bearing E0771 tumors were treated with PBS, control, CP (anti-CTLA4 and anti-PD-L1), RT, control + CP or RT + CP. *Created using BioRender.com. (**B**) Tumor volumes were measured every 2–3 days from the start of treatment until mice reached endpoint. Each line represents an individual mouse within the group. (**C**) Kaplan–Meier survival curves of each group. ** *p* < 0.001.

**Figure 7 ijms-22-04843-f007:**
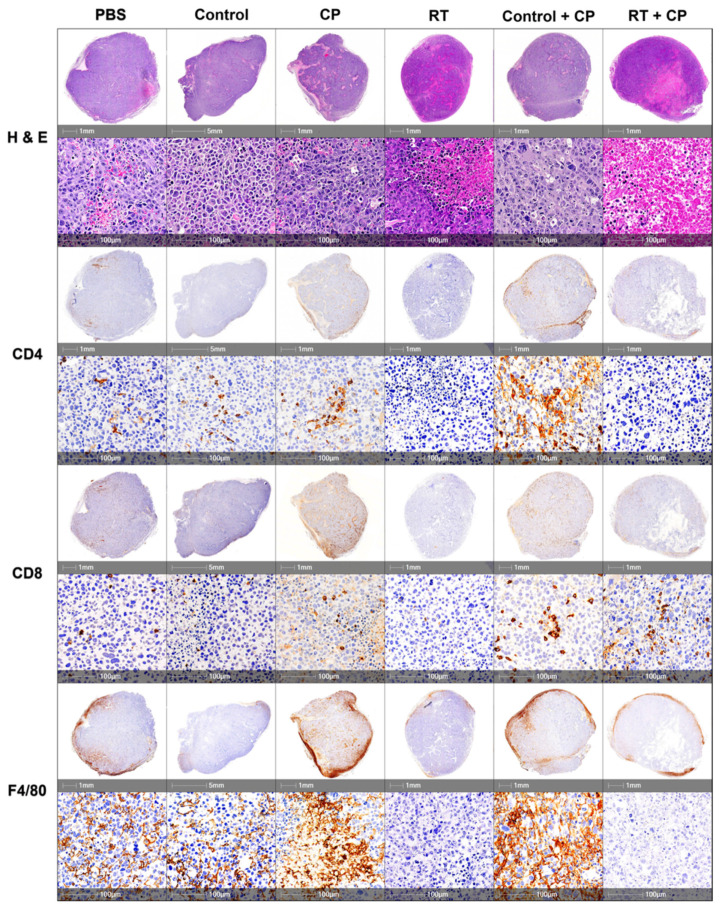
IHC analysis of immune cell infiltrates. C57/Bl6 mice bearing E0771 tumors were treated with PBS, control, CP, RT, control + CP, or RT + CP, and tumors were harvested on day 7. Tumors were sectioned and stained with H&E, CD4, CD8, and F4/80 for pathologic analysis. Each representative image shows a whole section of an individual tumor from the given group.

**Figure 8 ijms-22-04843-f008:**
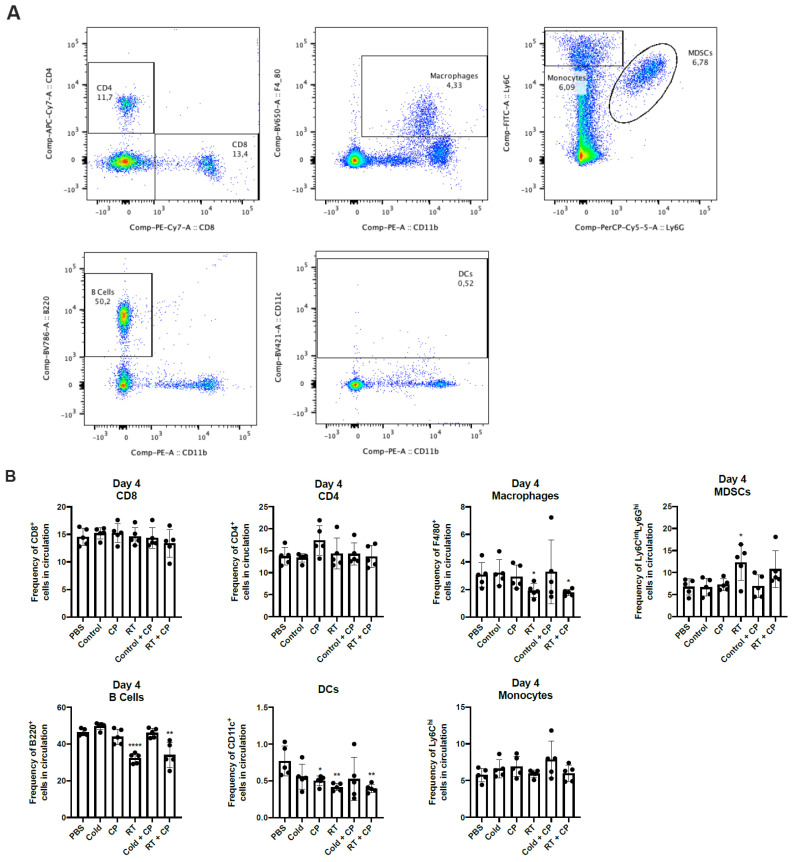
RT + CP decreases immunosuppressive MDSCs and B cells in the peripheral blood. C57/Bl6 mice bearing E0771 tumors were treated with either PBS, Control, CP, RT, Control + CP, or RT + CP. Blood was drawn on days 4 and 9 and analyzed via flow cytometry. (**A**) Representative flow plots showing the gating strategy for CD4^+^ T cells, CD8^+^ T cells, F4/80^+^ macrophages, Ly6C^hi^Ly6G^-^ monocytes, Ly6G^hi^Ly6C^int^ MDSCs, B220^+^ B cells, and CD11c^+^ DCs. (**B**) Bar plots showing the frequency of cells in circulation on day 4. (**C**) Bar plots showing the frequency of cells in circulation on day 9. * *p* < 0.05; ** *p* < 0.01; *** *p* < 0.001; **** *p* < 0.0001.

## Data Availability

All data is contained within this manuscript and within the Appendix A.

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
