# Peer review of "Combined Radionuclide Therapy and Immunotherapy for Treatment of Triple Negative Breast Cancer"

_ijms, 2021, doi:10.3390/ijms22094843_

Round 1

Reviewer 1 Report

An interesting article dealing with radionuclide therapy and immunotherapy combinations in mice, that could possibly lead to new therapeutic strategies. However, modifications appears necessary to increase its value

Some errors in figures numbers must be corrected. For example, figures 3 and 4 are used for 2 different figures, each. 

Reference 26 appears somewhat vague regarding the topics, a more orientated reference should be cited.

Institutional Review Board Statement is missing

Major comment: The main limitation is on the technical approach as the radionucleide administration is intratumoral, reducing the possibilities in many clinical situations. It sould be discussed in the discussion part of the manuscript. While the authors discuss the potential translatability of this model in L293-295, this route of administration clearly reduce the aforementionned translatability. A section evaluating the distribution of the RT after an IV administration would be of great interest in order to evaluate the exact interest of these intra-tumor administration.

Some experiments appears missing, to my point of view:

  • a biodistribution experiment withIV radionucleide administration, to evaluate the possible tumor tissue -targeting of their radionucleide-albumin agent (as discussed previously)
  • in the FACS expremiments, it is somewhat complex to evaluate the part of modifications of the different immune compartments linked to direct effect of CP and RT, and direct toxicity of these compounds. The authors should add informations on the global blood parameters (white blood count, total lymphocytes, monocytes and polynuclears) in order to try to capture these two aspects

Author Response

Reviewer #1 (Remarks to the Author):

An interesting article dealing with radionuclide therapy and immunotherapy combinations in mice, that could possibly lead to new therapeutic strategies. However, modifications appear necessary to increase its value

  1. Some errors in figures numbers must be corrected. For example, figures 3 and 4 are used for 2 different figures, each. 
  2. Reference 26 appears somewhat vague regarding the topics a more orientated reference should be cited.
  3. Institutional Review Board Statement is missing
  4. Major comment: The main limitation is on the technical approach as the radionuclide administration is intratumoral, reducing the possibilities in many clinical situations. It should be discussed in the discussion part of the manuscript. While the authors discuss the potential translatability of this model in L293-295, this route of administration clearly reduce the aforementioned translatability. A section evaluating the distribution of the RT after an IV administration would be of great interest in order to evaluate the exact interest of these intra-tumor administration.
  5. Some experiments appears missing, to my point of view:
  • a biodistribution experiment with IV radionuclide administration, to evaluate the possible tumor tissue -targeting of their radionuclide-albumin agent (as discussed previously)
  • in the FACS experiments, it is somewhat complex to evaluate the part of modifications of the different immune compartments linked to direct effect of CP and RT, and direct toxicity of these compounds. The authors should add information on the global blood parameters (white blood count, total lymphocytes, monocytes and polynuclears) in order to try to capture these two aspects

Author Response to Reviewer #1:

We would like to thank Reviewer #1 for their time reading our manuscript and providing feedback to improve upon our work.

  1. We apologize for the confusion, but the figures are “Figure 3”, “Figure 4”, “Figure S3” (Supplemental) and “Figure S4” (Supplemental).
  2. We have left reference 26 but added an additional reference (27) that further supports the statement in question (Line 220).
  3. The reviewer can find the institutional review board statement in the methods section “In Vivo Therapy Experiments” (Lines 387-390).
  4. We thank the reviewer for addressing these logistical issues with clinical administration of our therapeutic agent. There are many radiotherapies available to be utilized in i.v. systemic delivery. However, as outlined in the results (Lines 93-96) and the discussion (Lines 304-309) we have stated that our intention was to specifically develop a radiotherapy platform for use with intratumoral delivery of the radioactive isotope. This is in an effort to combat the widespread toxicity to non-target tissue that is seen with systemic administration of radiotherapeutics. Indeed, while i.v. administration is undoubtedly favourable for most therapy platforms, we have seen a recent insurgence of intratumoral therapeutics in the clinic, validating this approach for many types of cancer [1–3].

5a. We have previously done the experiment requested but chose not to initially include the data as it was a non-targeted study, and the results were poor (as expected). We have now added this data into the manuscript in Figure S2b and Table S2 (Lines 134-139, 148-155 and 179-180).

5b. We agree with Reviewer #1 that flow cytometry analysis is challenging to directly link to the effects of CP and RT, due to the inherent complexity of the immune system and the many different cellular types. We have utilized this technique to gain a baseline view of the immune population changes within the mice, but ideally would have looked at the populations directly within the tumor itself (though we were unable to do so by the nature of the tumors being radioactive and unable to be used on our cytometer without contaminating the machine and facilities). We have added in some additional populations to Figure 8, as per Reviewer #1’s suggestion. Specifically, we have added in frequencies of B cells, monocytes and dendritic cells to offer a more global picture of the immune cell levels in the peripheral blood (Lines 276-282 and 288-295).

Reviewer 2 Report

The paper describes a semi-novel therapeutic approach to triple negative breast cancer in a mouse model. The experimental approach is based on co-stimulation blockade (checkpoint blockade) combined with localized radio-therapy, which is not novel on itself, but the Authors approach this combination using a novel perspective.

Results show that this approach is effective in prolonging mice survival.

I think the paper would be improved by adding the following information:

  • it is known that co-stimulation blockade has significant safety signals in humans and treatment is complicated by significant side effects that have limited the clinical use of these drugs. It is recommended that information on the side effects of this drug combination is provided as part of this manuscript. The Authors mention that the radiotherapy is well tolerated, but i could not find any information on the potential side effects of the drug combination. Although the survival time was limited, this information could be helpful on the overall potential translation of this protocol to humans.
  • co-stimulation blockade has an important impact on the generation of T regulatory cells both systemically and within the tumor micro-environment. These cells have a very important role on the fate of the tumor and how the immune system will react to the treatment. I would recommend to the Authors to address this issue in the current manuscript, which would improve the understanding on the mechanisms of action of the therapeutic approach. The presence of radiotherapy could have a significant impact on the Tregs sub-populations and describing these changes would add significantly to the scientific value of this paper

Author Response

Reviewer #2 Remarks to the Author):

The paper describes a semi-novel therapeutic approach to triple negative breast cancer in a mouse model. The experimental approach is based on co-stimulation blockade (checkpoint blockade) combined with localized radiotherapy, which is not novel in itself, but the Authors approach this combination using a novel perspective.

Results show that this approach is effective in prolonging mice survival.

I think the paper would be improved by adding the following information:

  1. It is known that co-stimulation blockade has significant safety signals in humans and treatment is complicated by significant side effects that have limited the clinical use of these drugs. It is recommended that information on the side effects of this drug combination is provided as part of this manuscript. The Authors mention that the radiotherapy is well tolerated, but I could not find any information on the potential side effects of the drug combination. Although the survival time was limited, this information could be helpful on the overall potential translation of this protocol to humans.

  1. Co-stimulation blockade has an important impact on the generation of T regulatory cells both systemically and within the tumor microenvironment. These cells have a very important role on the fate of the tumor and how the immune system will react to the treatment. I would recommend to the Authors to address this issue in the current manuscript, which would improve the understanding on the mechanisms of action of the therapeutic approach. The presence of radiotherapy could have a significant impact on the Tregs sub-populations and describing these changes would add significantly to the scientific value of this paper

Author Response to Reviewer #2:

The authors would like to thank Reviewer #2 for taking the time to read our manuscript and provide useful feedback that will improve the depth and significance of the work.

  1. We agree with this clinical tolerability issue that Reviewer #2 has addressed. As such, we have added pertinent information regarding the regimen tolerability in the mice to address this (Line 230 and 318-320).
  2. We agree with the reviewer that Tregs may be playing an important role in our therapy. We have addressed this point in Lines 260 and 328-334.

Round 2

Reviewer 1 Report

The authors answered in a satisfactory way to the comments.

No additionnal comment.